# Modulator Effect of AT1 Receptor Knockdown on THP-1 Macrophage Proinflammatory Activity

**DOI:** 10.3390/biology13060382

**Published:** 2024-05-26

**Authors:** Lourdes Nallely Acevedo-Villavicencio, Carlos Enrique López-Luna, Juan Castillo-Cruz, Rocío Alejandra Gutiérrez-Rojas, Iris Selene Paredes-González, Santiago Villafaña, Fengyang Huang, Cruz Vargas-De-León, Rodrigo Romero-Nava, Karla Aidee Aguayo-Cerón

**Affiliations:** 1Escuela Superior de Medicina, Instituto Politécnico Nacional, Sección de Estudios de Posgrado e Investigación, Ciudad de México 11340, Mexico; dranallelyacevedo0611@gmail.com (L.N.A.-V.); jimmyubermensch@live.com (C.E.L.-L.); juancast0508@gmail.com (J.C.-C.); santiagovr1@gmail.com (S.V.); leoncruz82@yahoo.com.mx (C.V.-D.-L.); 2Escuela Nacional de Ciencias Biológicas, Instituto Politécnico Nacional, Ciudad de Mexico 11340, Mexico; ross.grojas.22@gmail.com; 3Instituto de Investigaciones Biomédicas, Departamento de Inmunología, Universidad Autónoma de México, Ciudad de México 70228, Mexico; iris_selene142@hotmail.com; 4Laboratorio de Investigación en Obesidad y Asma, Hospital Infantil de México Federico Gómez, Ciudad de Mexico 06720, Mexico; huangfengyang@gmail.com; 5División de Investigación, Hospital Juárez de Mexico, Mexico City 07760, Mexico

**Keywords:** telmisartan, AT_1_R siRNA, macrophage, proinflammatory cytokines

## Abstract

**Simple Summary:**

The macrophage is a type of cell that belongs to the immunological system, but it is involved in different activities in other systems due to their main role in the development of inflammatory response. The inflammation process is considered a state of the organism in which proinflammatory cytokines that are synthesized by macrophage are released and induce the synthesis of other molecular components that maintain the process through the binding of the specific receptors. Macrophages express different kinds of receptors, one of them is Ang II receptor, which participates in the modulation of proinflammatory cytokines synthesis in macrophages by using antagonists. In this study, we use the gene silencing of the AT_1_ receptor as a strategy to modulate the proinflammatory activity in macrophage and we found that the use of AT_1_ receptor siRNA is an alternative to modulate the macrophage proinflammatory activity, so it could be an option to control the inflammatory process in cardiac diseases.

**Abstract:**

Currently, it is known that angiotensin II (AngII) induces inflammation, and an AT_1_R blockade has anti-inflammatory effects. The use of an AT_1_ receptor antagonist promotes the inhibition of the secretion of multiple proinflammatory cytokines in macrophages, as well as a decrease in the concentration of reactive oxygen species. The aim of this study was to determine the effect of AT_1_ receptor gene silencing on the modulation of cytokines (e.g., IL-1β, TNF-α, and IL-10) in THP-1 macrophages and the relation to the gene expression of NF-κB. Materials and Methods: We evaluated the gene expression of PPAR-γ in THP-1 macrophages using PMA (60 ng/mL). For the silencing, cells were incubated with the siRNA for 72 h and telmisartan (10 µM) was added to the medium for 24 h. After that, cells were incubated during 1 and 24 h, respectively, with Ang II (1 µM). The gene expression levels of AT_1_R, NF-κB, and cytokines (IL-1β, TNF-α, and IL-10) were measured by RT-qPCR. Results: We observed that silencing of the AT_1_ receptor causes a decrease in the expression of mRNA of proinflammatory cytokines (IL-1β and TNF-α), NF-κB, and PPAR-γ. Conclusions: We conclude that AT_1_R gene silencing is an alternative to modulating the production of proinflammatory cytokines such as TNF-α and IL-1β via NF-κB in macrophages and having high blood pressure decrease.

## 1. Introduction

Obesity and metabolic disease are characterized by low-grade chronic inflammation in adipose tissue, as well as the accumulation of macrophages and T cells and increased proinflammatory cytokine levels [1]. This low-grade inflammation could lead to the development of type 2 diabetes mellitus (T2DM), hypertension, metabolic syndrome (MS), insulin resistance (IR), and other associated pathologies [2,3]. In adipose tissue, there is a high level of infiltration of CD8+ T cells, which is responsible for the development of adipose tissue inflammation, followed by macrophage accumulation [4]. Macrophages normally infiltrate the adipose tissue, triggering an obesogenic environment that provokes the deterioration of homeostasis due to the rebalancing of the M1 and M2 macrophage populations [5]. An increase in the M1 phenotype in adipose tissue favors the development of insulin resistance and inflammation and triggers the secretion of proinflammatory cytokines [6]. However, there have been reports associating the renin–angiotensin system with the modulation of the differentiation process in adipocytes [7]. Angiotensin I (Ang I) is a peptide formed by angiotensinogen (Agt) and processed by Angiotensin II (Ang II) [8].

Ang II is considered the major effector peptide of the renin–angiotensin system (RAS). It can enter the cell using Angiotensin II receptor type 1 (AT_1_R) or be stored in the intracellular compartments. It increases vascular permeability and activates inflammatory and antimicrobial responses by increasing the production of reactive oxygen species that produce oxidative stress [9]. Moreover, it can activate Toll-like receptor 4 (TLR4) receptors by regulating the expression of multiple substances, including growth factors, cytokines, chemokines, and adhesion molecules, which are involved in cell growth and proliferation [10]. The expression of Ang II receptors on macrophages, either in vitro or in vivo, suggests that Ang II synthetized by macrophages acts through an autocrine/paracrine mechanism [11]; other in vitro studies described regulation of the effect of AngII by C-reactive protein (CRP) and upregulation of AT_1_R in vascular smooth muscle cells [12]. Several studies have shown that the use of an AT_1_R antagonist promotes inhibition of the secretion of multiple proinflammatory cytokines in macrophages, as well as a decrease in the concentration of reactive oxygen species [13], through the activation of certain transcription factors such as c-Jun N-terminal kinase (JNK) [14], NF-κB, and activator protein (AP-1), which are involved in the release of proinflammatory cytokines mediated by Ang II [15].

Ang II activates NF-κB, which, in turn, increases inflammation-mediated tissue damage [16]. In macrophages, this peptide, through the AT_1_R, could modulate the production of different cytokines, such as tumor necrosis factor alpha (TNF-α), interleukin-1 beta (IL-1β), IL-6, and IL-10, and its antagonism produces anti-inflammatory effects [17,18,19]. According to Wu et al. [18], Ang II may induce a M1 phenotype in macrophages using the NF-κB pathway. Telmisartan is a drug belonging to the Ang II receptor antagonist (ARB) family that is used in the management of hypertension. It is a competitive AT_1_R antagonist that lacks intrinsic activity at the AT_1_R [20,21]. The main effects of telmisartan are vasoconstriction, aldosterone release, and lowering of systemic blood pressure [21]. Additionally, there have been recent studies that have described other properties of telmisartan as an insulin sensitivity modulator via peroxisome proliferator-activated receptor γ (PPARγ) [22,23].

In the present study, we used Ang II as a stimulus that promotes the synthesis of proinflammatory cytokines. We also used telmisartan as an antagonist that inhibits the intrinsic activity of the AT_1_R to decrease the synthesis of TNF-α and IL-1β. The aim of this study was to determine the effect of AT_1_R gene silencing on the modulation of proinflammatory cytokines and the possible interplay between the AT_1_R signaling pathway and NF-κB inhibition.

## 2. Material and Methods

### 2.1. Reagents

The THP-1 cell line was obtained from the ATCC (TIB-202; human monocyte cell line). Dulbecco’s Modified Eagle Medium/Nutrient Mixture F12 (DMEM/F12), Fetal Bovine Serum (HyClone, Logan, UT, USA), phorbol 12-myristate 13-acetate, 12-O-tetradecanoylphorbol 13-acetate (PMA), and dimethyl sulfoxide (DMSO) were purchased from Caledon Laboratories (Chalk River, Canada). 1-(4,5-Dimethylthiazol-2-yl)-3,5-diphenylformazan and thiazolyl blue formazan (MTT) were obtained from Sigma-Aldrich (Burlington, MA, USA). Phosphate-buffered saline (PBS) NaCl (137 mml/L), KCl (2.7 mmol/L), Na_2_HPO_4_ (10 mmol/L), and KH_2_PO_4_ (1.8 mmol/L], pH 7.0) were also obtained from Sigma-Aldrich (Burlington, MA, USA). TRIzol reagent was purchased from Invitrogen; Ang II human (AngII) and telmisartan were obtained from Sigma-Aldrich; and AT_1_R siRNA was obtained from the Dharmacon TM reagent.

### 2.2. Proliferation and Differentiation of THP-1

THP-1 cells were cultured in 25 cm^2^ flasks in DMEM/F12 supplemented with 10% FBS and 1% penicillin/streptomycin at 37 °C, 5% CO_2_, in a humidified atmosphere. To differentiate THP-1 cells into macrophages, cells were seed in a 24-well plate (3 × 10^5^ cells per well) in DMEM/F12 and incubated with 60 ng/mL of PMA for 48 h. The medium was changed every 2 days.

### 2.3. Culture Conditions with Treatments

Cells were plated in 24-well plates at 2 × 10^5^ cells per well in 250 µL of DMEM/F12 culture medium supplemented with 10% FBS. The cells were incubated with telmisartan (10 mM) for 24 h after Ang II (1 µM) was added to the medium, and the cells were incubated for 1 or 24 h.

### 2.4. Small Interfering RNA (siRNA)

The siRNA for AT_1_R in humans was designed using siRNA WizardTM (InvivoGen). This algorithm allows the selection of a siRNA specific to the mRNA that encodes the AT_1_R protein according to the thermodynamic criteria that are necessary for gene silencing [24]. The mRNA secondary structure of the AT_1_ receptor and AT_1_ siRNA hybridization site was obtained using RNA Fold software (http://rna.tbi.univie.ac.at/cgi-bin/RNAWebSuite/RNAfold.cgi, accessed on 19 May 2024) (Appendix A). Cells were plated in 24-well plates at 2 × 10^5^ cells per well in 250 µL of DMEM/F12 culture medium supplemented with 10% FBS. Cells were transfected for 72 h with siRNA (500 ng) using Lipofectamine 2000 as a vehicle in Opti-MEM, according to the manufacturer’s instructions. Then, the medium was replaced, and cells were incubated for 1 or 24 h with Ang II (1 µM).

### 2.5. RNA Extraction and Genetic Expression of Cytokines

Gene expression was evaluated by RT-qPCR, and the primers for mRNA amplification were designed by the Universal Probe Library Assay Design Center (LifeScience, Roche, https://primers.neoformit.com/) and synthetized by T4-oligo (Table 1). THP-1 cells were plated in 24-well plates at 2 × 10^5^ cells per well and differentiated into macrophages. Then, RNA was extracted from the cells by using TRIzol reagent, according to the manufacturer’s instructions. The concentration and purity of RNA were evaluated using a nanophotometer (Implen Inc. Estlake Village, CA, USA) at the wavelengths of 260/280 nm and 260/230 nm, respectively. A ratio of 1.8 to 2.2 indicated acceptable purity for this study. The cDNA was prepared using M-MLV reverse transcriptase (Thermo Fisher Scientific, Waltham, MA, USA) and 500 ng of total RNA. The RT-qPCR was performed using a mix of SYBR green qPCR master mix (Thermo Scientific, Waltham, MA, USA) and the appropriate primers. The RT-qPCR conditions were initial denaturation at 95 °C; 45 cycles of 15 s at 90 °C, 30 s at 60 °C, and 15 s at 72 °C; and cooling 300 s at 40 °C.

### 2.6. Enzyme-Linked Immunosorbent Assay (ELISA)

IL-1β and TNF-α cytokine production were evaluated by an ELISA kit (R&D Systems, Minneapolis, MN, USA), according to the manufacturer’s instructions. We collected the medium after 1 and 24 h. The absorbance value (OD 450 nm) was measured with the Benchmark Plus Microplate Reader (Bio-Rad, Hercules, CA, USA).

### 2.7. Statistical Analysis

The results were expressed as the mean ± standard error (SE). The relative changes in gene expression levels were calculated using the 2^−∆∆*C*^_T_ method [25]. Differences between groups were determined by unidirectional analysis of variance (ANOVA) with Tukey´s post-hoc test. For our analysis, we used GraphPad Prism 7.0. The threshold for significance was set at *p* < 0.05.

## 3. Results

### 3.1. The Effect of AT_1_R siRNA

As shown in Figure 1, the stimulus with Ang II (1 µM) after 1 h (Figure 1A) increased the gene expression levels of AT_1_R (9-fold) compared with the control group. By contrast, we observed that in the group preincubated with telmisartan (10 mM) for 24 h, then stimulated with Ang II for 1 h, the drug induced a significant decrease in AT_1_R gene expression (Figure 1A) compared with the group stimulated with Ang II, and we did not find a significant difference compared with the control group of cells that did not receive any kind of treatment or stimulus. Finally, in the group transfected with AT_1_R siRNA, we observed a result like that in the group treated with telmisartan, and additionally, in the stimulus with Ang II for 1 h (Figure 1A), the expression of AT_1_R did not change in comparison with the control group, but we observed a significant decrease compared with the Ang II group (Figure 1A).

According to the results of 24 h of Ang II stimulus (Figure 1B), we observed overexpression of the AT_1_R (30-fold) in macrophages. In the group treated with telmisartan (10 mM), the expression of the AT_1_R decreased in comparison with the Ang II-stimulated group (Figure 1B), but in the group that was incubated for 24 h with Ang II, the expression of AT_1_R increased in comparison with the group that was incubated for only 1 h with Ang II.

Finally, in the group transfected with AT_1_R siRNA, the results showed that AT_1_R gene expression decreased in comparison with the group stimulated with Ang II (Figure 1B), and we observed that compared with the group transfected with siRNA and stimulated for 1 h with Ang II, AT_1_R expression decreased in the group transfected with AT_1_R siRNA and stimulated with Ang II for 24 h (Figure 1B).

### 3.2. siRNA AT_1_R Modulates NF-κB Expression in THP-1 Cells

The results showed that NF-κB gene expression decreased in the group that was stimulated with Ang II (1 µM) for 1 h (Figure 2A) compared to the control group, and in the groups with telmisartan and AT_1_R siRNA, we observed that NF-κB gene expression decreased compared to the Ang II group (Figure 2A).

On the other hand, we observed that in cells stimulated with Ang II (1 µM) for 24 h, NF-κB gene expression increased significantly in comparison with the control group (Figure 2B). Finally, in the group treated with telmisartan and AT_1_R siRNA, NF-κB gene expression decreased in comparison with the Ang II-treated group (Figure 2B).

### 3.3. AT_1_R siRNA Changes the Gene Expression of Cytokines in Macrophages

The results showed that in macrophages stimulated with Ang II (1 µM) for 1 h, the gene expression of TNF-α (Figure 3A) increased (5-fold) in comparison with the control group, and we observed that the telmisartan-treated cells (24 h before Ang II stimulation) showed a significant decrease in TNF-α gene expression (Figure 3A) when compared with the group stimulated only with Ang II. Finally, we observed similar results in the group transfected with AT_1_R siRNA and in the Ang II-stimulated group (Figure 3A). Regarding the cells stimulated for 24 h with Ang II, we observed that AT_1_R expression increases significantly in the group stimulated with Ang II compared with the control group; however, we did not observe a difference in AT_1_R gene expression in the group treated with telmisartan before the Ang II stimulus compared with the group stimulated with Ang II alone, but we observed that AT_1_R gene expression decreased in the AT_1_R siRNA group (Figure 3B).

The gene expression of IL-1β showed similar results in the group stimulated with Ang II for 1 h (Figure 3C) and 24 h (Figure 3D); the expression of this cytokine increased in the group stimulated with Ang II compared with the control group, but in other groups (telmisartan and AT_1_R siRNA), its expression decreased compared with the Ang II group (Figure 3C,D).

The evaluation of IL-10 gene expression did not show changes in the Ang II-stimulated group compared with the control group at 1 h and 24 h (Figure 3E,F). Finally, the results showed no difference in the telmisartan group compared with the Ang II-stimulated group (Figure 3E). By contrast, we observed an increase in the expression of IL-10 in the AT_1_R siRNA group compared to the group stimulated with Ang II for 1 h (Figure 3E). On the other hand, in the group stimulated with Ang II for 24 h, we did not observe significant differences in the expression of IL-10 compared with any of the other groups (Figure 3F).

### 3.4. PPAR-γ Gene Expression in Macrophages

We evaluated the expression of PPAR-γ in macrophages, and the groups stimulated with Ang II for 1 h (Figure 4A) and 24 h (Figure 4B) showed similar results: In the group stimulated with Ang II, PPAR-γ expression increased in comparison with the control group (Figure 4A,B). By contrast, in the group treated with telmisartan and transfected with AT_1_R siRNA, PPAR-γ expression in macrophages decreased in comparison with the group stimulated with Ang II for 1 h (Figure 4A) and 24 h (Figure 4B).

### 3.5. AT_1_ siRNA Inhibits Ang II-Induced Inflammation in THP-1 Macrophages by Regulating TNF-α and IL-1β Secretion

To investigate how gene silencing of AT_1_ modulates Ang II-induced inflammation, we measured the protein secreted. The protein levels of TNF-α and IL-1β increased 1 h and 24 h after the Ang II stimulus in macrophages compared with the control group (Figure 5). In the group transfected with AT_1_R siRNA, we observed a significant decrease compared with the group stimulated with Ang II. These findings were similar 1 and 24 h after angiotensin stimulation. On the other hand, we used an AT_1_ antagonist as a positive control to evaluate the effect of AT_1_ siRNA on proinflammatory cytokine (TNF-α and IL-1β) release, and we observed similar results for AT_1_ siRNA (Figure 5).

## 4. Discussion

The renin–angiotensin–aldosterone system is involved in the modulation of many functions related to vascular function: injury, disfunction, remodeling, and inflammation, and the macrophage is one of the most important cells that participates in and contributes to the maintenance of balance in the endothelium, but its role in the development of inflammation is associated with the synthesis of chemokines and cytokines [11]. On the other hand, Ang II is a peptide related to vascular injury that acts as a mediator by binding to AT_1_R and AT_2_R [11,24,25,26]. Some of the activities of Ang II are increasing blood pressure, forming NO by stimulating NOS, and changing the phenotype of vascular smooth muscle cells (VSMCs) [27,28].

The aim of this study was to demonstrate silencing of AT_1_R in a human macrophage cell line as an alternative pathway for modulating inflammatory development and even Ang II stimuli. We evaluated changes in the gene expression of the main cytokines involved in the inflammatory environment, such as TNF-α and IL-1β, in addition to the expression of IL-10, a cytokine that is considered an anti-inflammatory factor synthetized by macrophages. In addition, we evaluated the expression of NF-κB, a transcription factor involved in the synthesis of these cytokines. Finally, we evaluated the expression of PPAR-γ because it is a molecule associated with beneficial effects on hypertension. The novel findings of this study are as follows: (1) telmisartan is less effective than AT_1_R siRNA as a blocker of Ang II in macrophages; (2) AT_1_R siRNA is an alternative to modulating inflammatory activity in macrophages by decreasing NF-κB gene expression; and (3) AT_1_R siRNA modifies proinflammatory cytokine (IL-1β and TNF-α) production and increases PPAR-γ gene expression in macrophages. A macrophage is a cell that plays a key role in the inflammation process and participates in inflammation via the activation of the RAS pathways due to the activation of AT_1_R by Ang II [10].

First, our results showed that Ang II stimulation raised the levels of AT_1_R mRNA in macrophages, so we can confirm that the concentration and duration of the stimulus function in the control of the stimulation of AT_1_R in macrophages. However, we observed that telmisartan only decreased the gene expression of AT_1_R in the group stimulated with Ang II for 1 h; in 24 h, the effect of the drug was lower. In addition, in AT_1_R knockout macrophages, the levels of AT_1_R mRNA decreased during both periods of Ang II stimulation, and the results suggest that in silencing macrophages, the effect of this molecular method depends on the concentration of the ligand. NF-κB is a transcription factor, and its activation is related to the synthesis of inflammatory cytokines in macrophages [29]. Ang II plays a crucial role in the activation of NF-κB in T lymphocytes, monocytes, and macrophages and promotes the synthesis of IL-1β and TNF-α, which are cytokines involved in inflammation process [30,31]. Such activation can occur via the IKK complex, giving rise to the phosphorylation of IκBα and p65, followed by nuclear translocation of the active NF-κB complex, which acts as a transcription factor [31,32,33]. According to the work of Li et al. [34], the role of salvianolic acid (SSA), a component of *Salvia miltiorrhiza*, was studied on the expression of NF-κB in murine macrophages using Ang II (1 µM) for 12 h, finding that Ang II increases the expression of NF-κB by binding with AT_1_R that induces a proinflammatory effect through the downstream activation of the intracellular signaling cascade of NF-κB activation. Our results confirm that Ang II binding to AT_1_R upregulates NF-κB in a time-dependent manner because we observed that 1 h of stimulus with Ang II could not increase the expression of this transcription factor in macrophages, but 24 h of stimulus with Ang II increased the expression of NF-κB, the same effect we found with telmisartan, so these results confirm the beneficial effects of telmisartan in the modulation of inflammation induced by macrophages. We observed that the effect of AT_1_R siRNA lasted longer than that of the drug, so the modulation of inflammation by decreasing inflammatory cytokine synthesis by macrophages is stronger in the case of AT_1_R knockdown.

It has been described that AT_1_R blockers not only decrease the expression of chemokines that avoid macrophage infiltration but also contribute to reduce inflammation by modulating the macrophage phenotype [35,36]. Aki et al., in 2010 [37], described the effect of olmesartan in the modulation of the macrophage phenotype; their results concluded that this drug induces a M2 phenotype, and they suggested that by switching the macrophage phenotype, the level of inflammation can be reduced through the inflammatory cytokines declining in quantity. However, there is little information about the relationship between the production of IL-1β and the activation of AT_1_R in macrophages. In this study, we evaluated the expression of IL-1β after a stimulus with Ang II for 1 h and 24 h, and the AT_1_R siRNA decreased the gene expression level of IL-1β, although the cells received a stimulus with Ang II. This result is related to the decreasing gene expression of NF-κB, a transcription factor that improves the synthesis of IL-1β. However, it is interesting that the cells with the telmisartan treatment did not decrease the expression of the transcription factor when the cells were stimulated with Ang II for 24 h, but in silenced cells, the result of a decreasing gene expression of NF-κB are very similar to the result of IL-1β gene expression. Li et al. [38] demonstrated that telmisartan treatment decreased apoptosis and inflammation in pancreatic cells, so our results showed that siRNA has an effect not only in the decline of IL-1β but also in the reduction of the transcription factor in macrophages and it could reduce not only the inflammation associated with diabetes but also the inflammation associated with cardiovascular disease because, in previous information, it is described that Ang II is related to the M1 phenotype and the abundance of the M1 and M2 populations determines atherosclerotic plaque development [39].

It is well known that infiltrated macrophages in adipose and cardiac tissue are related to insulin resistance [40], specifically the M1 phenotype, which secretes TNF-α. Additionally, this cytokine is a main component of inflammation; due to this relationship, hypertension and cardiovascular disease are associated with the M1 phenotype. In our results, we found that AT_1_R siRNA significantly attenuates the expression of TNF-α despite the stimulus with Ang II for 1 or 24 h, so this result suggests that the silencing of AT_1_R in macrophages could decrease systemic inflammation and prevent hypertrophy, according to previous studies that described the relationship between the TNF-α and hypertrophy development via NF-κB [41]. Moreover, we confirm the effect not only in the decreasing gene expression of both crucially important inflammatory cytokines related to the proinflammatory function of macrophages: IL-1β and TNF-α, but also, we observed a decrease in the release of these cytokines (protein expression quantification) due to the effect of AT_1_ siRNA on NF-κB gene expression.

It has been described that there is an independent system that regulates RAS in the brain; IL-10 is an anti-inflammatory cytokine related to the inhibition of the development of neurodegenerative disease [42]; and the treatment with losartan could reduce the level of oxidative stress on the brain [43]. Additionally, there is evidence that blood pressure is reduced in mice treated with IL-10 [44]. Finally, we evaluated IL-10, an anti-inflammatory cytokine, and we did not find significant differences between our groups regarding the gene expression level of this cytokine. Our results suggest that it is not enough to block AT_1_R to decrease the level of inflammation, but it is also necessary to stimulate AT_2_R to increase IL-10 production, as described in previous studies [45].

Some studies have shown that the effect of ARA II is temporary, so it is necessary to administer it several times to maintain its antagonistic action on macrophages [46]. Iwashita et al. [47] demonstrated that valsartan, an Ang II receptor blocker, similar to telmisartan, modulates the production of IL-1β, IL-6, and TNF-α without using the PPAR-γ or AT_1_R pathway; a possible mechanism is through the communication between adipocytes and macrophages. On the other hand, there are some reports that have described that losartan inhibits macrophage apoptosis through the MAPK pathway [48]. In our results, we observed that the telmisartan (10 µM) treatment for 24 h decreased the gene expression of PPAR-γ. This result matches with previous studies that described the decrease of PPAR-γ gene expression evaluated by RT-qPCR in cell cultures of THP-1 macrophages [49]. This result suggests that silencing of AT_1_R could help improve insulin resistance through the modulation of macrophage activity. Additionally, there is evidence that suggests that the modulation of Ang II action in macrophages prevent or avoid cardiac remodeling, so using siRNA AT_1_ could be a specific therapy for cardiac affection, and it could be an alternative treatment without using antagonists with a limited activity that, compared to the knockdown AT_1_ by siRNA, has an alternative activity that promotes the AT2 receptor and stimulates the functions of this receptor.

Limitations of our study include the evaluation of AT_2_R gene expression and other factors of the AT_1_ and AT_2_ receptor pathways, as well as other cytokines and chemokines that allow us to determine the phenotype of macrophages because macrophage M1 polarization and adhesion trigger the endothelial cell inflammatory response.

## 5. Conclusions

We conclude that AT_1_R gene silencing is an alternative to modulating the production and secretion of proinflammatory cytokines, such as TNF-α and IL-1β, via NF-κB in macrophages in spite of stimulation with Ang II. On the other hand, the results show that this AT_1_ knockdown also modulates the expression of the IL-10 messenger, possibly due to the participation of the Ang II-dependent AT_2_ receptor, and we observed that the effect of the silencing is time-dependent. Moreover, the AT_1_R siRNA has an effect on the gene expression of PPAR-γ in macrophages. These findings have potentially important implications for knowledge of activated pathways via AT_1_R in macrophages.

## Figures and Tables

**Figure 1 biology-13-00382-f001:**
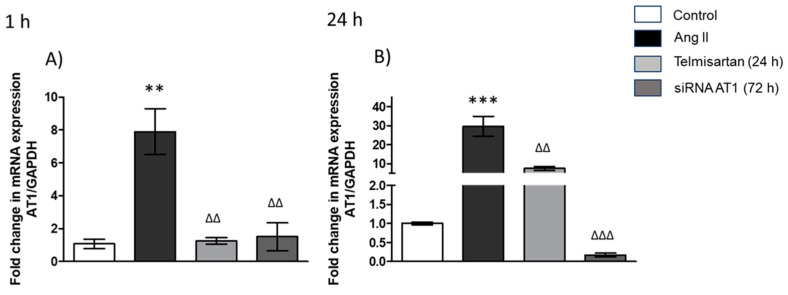
AT_1_R gene expression. (**A**) Macrophages stimulated with Ang II (1 µM) for 1 h; (**B**) macrophages stimulated with Ang II (1 µM) for 24 h. mRNA levels were examined using real-time RT-qPCR. Control = cells with neither treatment nor stimulus; Ang II = cell with Ang II stimulus; telmisartan = cells incubated for 24 h with telmisartan (10 mM) + Ang II (1 µM); siRNA AT_1_R = cell with AT_1_R siRNA (72 h) + Ang II group (1 µM). Results are expressed as the means ± SEMs. ** *p* < 0.01 and *** *p* < 0.001 versus the control group. ^ΔΔ^
*p* < 0.01 and ^ΔΔΔ^
*p* < 0.001. One-way ANOVA, followed by Tukey´s post-hoc test.

**Figure 2 biology-13-00382-f002:**
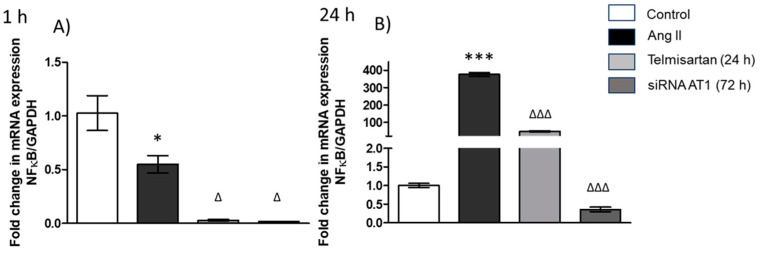
NF-κB gene expression. (**A**) Macrophages stimulated with Ang II (1 µM) for 1 h; (**B**) macrophages stimulated with Ang II (1 µM) for 24 h. mRNA levels were examined using real-time RT-qPCR. Control = cells with neither treatment nor stimulus; Ang II = cells with Ang II stimulus; telmisartan = cells incubated for 24 h with telmisartan (10 mM) + Ang II (1 µM); siRNA AT_1_R = cells transfected with AT_1_R siRNA (72 h) + stimulated with Ang II (1 µM). Results are expressed as the means ± SEMs. * *p* < 0.05 and *** *p* < 0.001 versus the control group. ^Δ^
*p* < 0.05 and ^ΔΔΔ^
*p* < 0.001. One-way ANOVA, followed by Tukey´s post-hoc test.

**Figure 3 biology-13-00382-f003:**
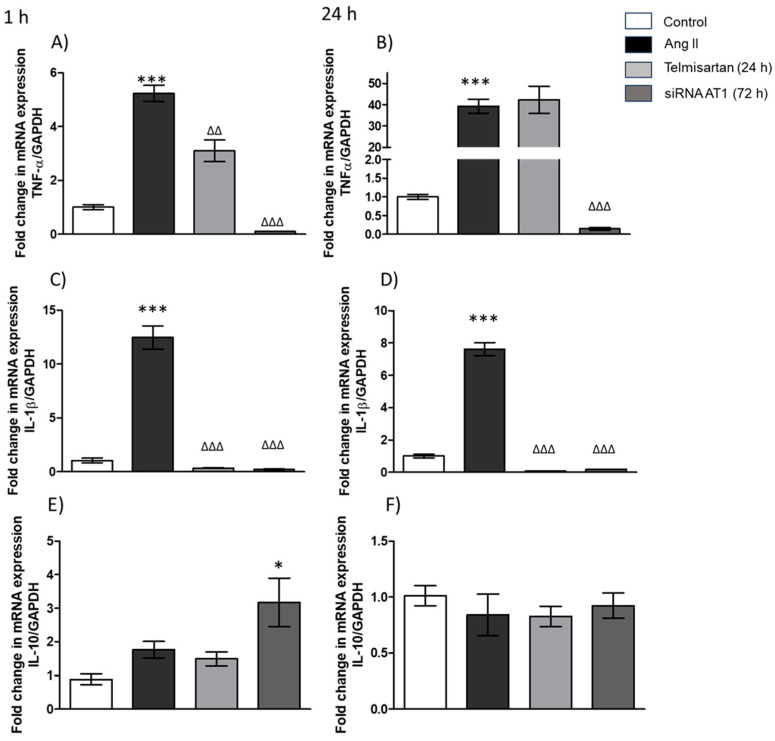
Cytokines gene expression in macrophages. (**A**) TNF-α gene expression in macrophages stimulated with Ang II (1 µM) for 1 h; (**B**) TNF-α gene expression in macrophages stimulated with Ang II (1 µM) for 24 h; (**C**) IL-1β gene expression in macrophages stimulated with Ang II (1 µM) for 1 h; (**D**) IL-1β gene expression in macrophages stimulated with Ang II (1 µM) for 24 h; (**E**) IL-10 gene expression in macrophages stimulated with Ang II (1 µM) for 1 h; (**F**) IL-10 gene expression in macrophages stimulated with Ang II (1 µM) for 24 h. mRNA levels were examined using real-time RT-qPCR. Control = cells with neither treatment nor stimulus; Ang II = cell with Ang II stimulus; telmisartan = cells incubated for 24 h with telmisartan (10 mM) + Ang II (1 µM); siRNA AT_1_R = cells with AT_1_R siRNA (72 h) + Ang II group (1 µM). Results are expressed as the means ± SEMs. * *p* < 0.05, *** *p* < 0.001 versus the control group. ^ΔΔ^
*p* < 0.01 and ^ΔΔΔ^
*p* < 0.001. One-way ANOVA, followed by Tukey´s post-hoc test.

**Figure 4 biology-13-00382-f004:**
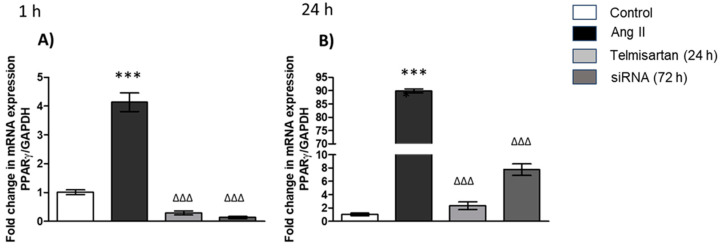
PPAR-γ gene expression in macrophages. (**A**) Macrophages stimulated with Ang II (1 µM) for 1 h; (**B**) macrophages stimulated with Ang II (1 µM) for 24 h. mRNA levels were examined using real-time RT-qPCR. Control = cells with neither treatment nor stimulus; Ang II = cells with Ang II stimulus; telmisartan = cells incubated for 24 h with telmisartan (10 mM) + Ang II (1 µM); siRNA AT_1_R = cells transfected with AT_1_R siRNA (72 h) + stimulated with Ang II (1 µM). Results are expressed as the means ± SEMs. *** *p* < 0.001 versus the control group. ^ΔΔΔ^
*p* < 0.001. One-way ANOVA, followed by Tukey´s post-hoc test.

**Figure 5 biology-13-00382-f005:**
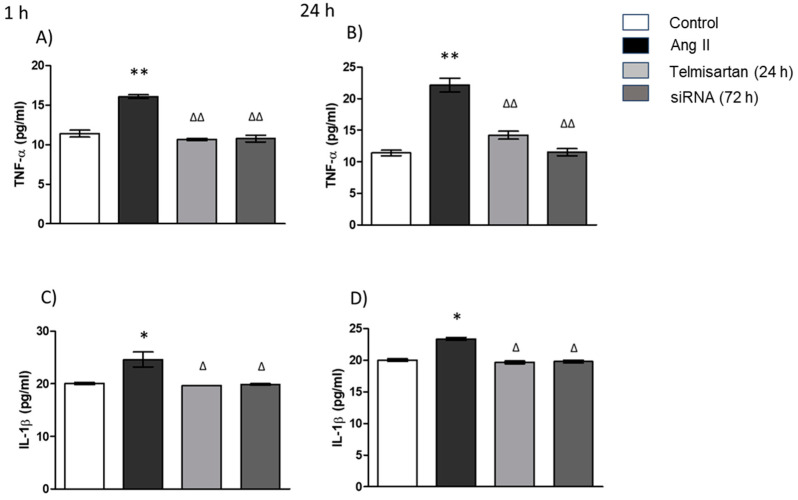
Cytokine protein expression in macrophages. (**A**) TNF-α protein expression in macro-phages stimulated with Ang II (1 µM) for 1 h; (**B**) TNF-α protein expression in macrophages stimulated with Ang II (1 µM) for 24 h; (**C**) IL-1β protein expression in macrophages stimulated with Ang II (1 µM) for 1 h; (**D**) IL-1β protein expression in macrophages stimulated with Ang II (1 µM) for 24 h. Protein levels were examined using ELISA. Control = cells with neither treatment nor stimulus; Ang II = cell with Ang II stimulus; telmisartan = cells incubated for 24 h with telmisartan (10 mM) + Ang II (1 µM); siRNA AT_1_R = cells with AT_1_R siRNA (72 h) + Ang II group (1 µM). Results are expressed as the means ± SEMs. * *p* < 0.05 and ** *p* < 0.01 versus the control group. ^Δ^
*p* < 0.05 and ^ΔΔ^
*p* < 0.01. One-way ANOVA, followed by Tukey´s post-hoc test.

**Table 1 biology-13-00382-t001:** Primer sequences used for real-time quantitative reverse transcriptase polymerase chain reaction (RT-qPCR).

Gene	5′ to 3′	Primers	Accession Number
AT_1_ receptor (AGTR1)	Sense	GCACTGGCTGACTTATGCTT	NM_000685.5
	Anti-sense	GTTGAAACTGACGCTGGCTG	
NF-κB	Sense	GGGCAGACCAGTGTCATTGA	NM_001077494.3
	Anti-sense	GTTGGTGAGGTTGACAACGC	
TNF-α	Sense	GTGCTTGTTCCTCAGCCTCT	NM_000594.4
	Anti-sense	TAGAGAGAGGTCCCTGGGGA	
IL-1β	Sense	AACGAGGCTTATGTGCACGA	NM_000576.3
	Anti-sense	TATCCTGTCCCTGGAGGTGG	
IL-10	Sense	CCAGTCTGAGAACAGCTGCA	NM_000572.3
	Anti-sense	TCCTCCAGCAAGGACTCCTT	
PPAR-γ	Sense	CCAGAAGCCTGCATTTCTGC	NM_001330615.4
	Anti-sense	CACGGAGCTGATCCCAAAGT	
GAPDH	Sense	CATCCTGGGCTACACTGAGC	NM_001256799.3
	Anti-sense	GTCAAAGGTGGAGGAGTGGG	

## Data Availability

The data presented in this study are available in the article.

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
