# Peer review of "Modulator Effect of AT1 Receptor Knockdown on THP-1 Macrophage Proinflammatory Activity"

_biology, 2024, doi:10.3390/biology13060382_

Round 1
Reviewer 1 Report
Comments and Suggestions for Authors
Although the findings that Angiotensin II receptor type 1 (AT1R) gene silencing is an alternative to modulate the production of proinflammatory cytokines such has TNF-α and IL-1β via NF-κB in macrophages, besides have high blood pressure decreased are interesting, numbers of points need clarifying and certain statements require further justification. These are given below.
<Point>
1 The authors entitled, “Modulator effect of AT1 receptor knockdown in macrophage proinflammatory activity” (lines 3-4). However, the authors used a human monocyte line THP-1 (Tsuchiya, S. et al. Int. J. Cancer 26, 171-176, 1980). Although the results are true in THP-1 cells, it is uncertain the results are true in macrophages. The authors should test other macrophages (cell lines) than THP-1 cells such as human HL-60, human MABS-1, human MOBS-1, mouseJ774A.1, mouse RAW264.7, and mouse BMA3.1A7. Alternatively, the authors change the title to “Modulator effect of AT1 receptor knockdown in THP-1 macrophage proinflammatory activity”.
2 “Caledon Laboratories, Canada” (lines 93-94) should be changed to “Caledon Laboratories, Chalk River, Canada”.
3 “Na2HPO4” (line 96) should be changed to “Na<sub>2</sub>HPO<sub>4</sub>”.
4 “KH2PO4” (line 96) should be changed to “KH<sub>2</sub>PO<sub>4</sub>”.
5 “Sigma-Aldrich, EE. UU.” (lines 96-97) should be changed to “Sigma-Aldrich, Burlington, MA”.
6 “25 cm2 flasks” (line 101) should be changed to “25 cm<sup>2</sup> flasks”.
7 “5% CO2” (line 102) should be changed to “5% CO<sub>2</sub>”.
8 “3 x 105 cells” (line 104) should be changed to “3 x 10<sup>5</sup> cells”.
9 “2 x105 cells” (line 108) should be changed to “2 x10<sup>5</sup> cells”.
10 “InvivoGen” (line 114) should be changed to “Invivogen”.
11 “2 x105 cells” (line 119) should be changed to “2 x10<sup>5</sup> cells”.
12 In Table 1, “NF-KB” should be changed to “NF-κB”.
13 “R&DSystems, Minneapolis, MN, USA” (lines 145-146) should be changed to “R&DSystems, Minneapolis, MN”.
14 “Bio-Rad, Calif, USA” (line 148) should be changed to “Bio-Rad, Hercules, CA”.
15 In Ref. 12, “Wang, C. H., Li, S. H., Weisel, R. D., Fedak, P. W., Dumont, A. S., Szmitko, P., ... & Verma, S.” should be changed to “Wang, C.H., Li, S.H., Weisel, R.D., Fedak, P.W., Dumont, A.S., Szmitko, P., Li, R.K., Mickle, D.A., Verma, S.”.
16 In Ref. 13, “Recent Patents on Cardiovascular Drug Discovery (Discontinued), 2(1), 23-27” should be changed to “Recent Pat Cardiovasc Drug Discov. 2007;2(1):23-27.”.
17 In Ref. 28, “Forrester, S. J., Booz, G. W., Sigmund, C. D., Coffman, T. M., Kawai, T., Rizzo, V., ... & Eguchi, S.” should be changed to “Forrester, S.J., Booz, G.W., Sigmund, C.D., Coffman, T.M., Kawai, T., Rizzo, V., Scalia, R., Eguchi, S.”.
18 In Ref. 39, “Aki, K., Shimizu, A., Masuda, Y., Kuwahara, N., Arai, T., Ishikawa, A., ... & Fukuda, Y.” should be changed to “Aki, K., Shimizu, A., Masuda, Y., Kuwahara, N., Arai, T., Ishikawa, A., Fujita, E., Mii, A., Natori, Y., Fukunaga, Y., Fukuda, Y.”.
19 In Ref. 40, “Journal of Diabetes Research, 2012” should be changed to “Exp. Diabetes Res. 2012;2012:618923.”.
20 In Ref. 49, “Valsartan, independientemente del receptor AT1 o PPAR, suprime la activación de macrófagos inducida por LPS y mejora la resistencia a la insulina en adipocitos cocultivados.” should be changed to “Valsartan, independently of AT1 receptor or PPARγ, suppresses LPS-induced macrophage activation and improves insulin resistance in cocultured adipocytes.”.
Author Response
Reviewer 1
- The authors entitled, “Modulator effect of AT1 receptor knockdown in macrophage proinflammatory activity” (lines 3-4). However, the authors used a human monocyte line THP-1 (Tsuchiya, S. et al. Int. J. Cancer 26, 171-176, 1980). Although the results are true in THP-1 cells, it is uncertain the results are true in macrophages. The authors should test other macrophages (cell lines) than THP-1 cells such as human HL-60, human MABS-1, human MOBS-1, mouseJ774A.1, mouse RAW264.7, and mouse BMA3.1A7. Alternatively, the authors change the title to “Modulator effect of AT1 receptor knockdown in THP-1 macrophage proinflammatory activity”.
Answer: thanks for the comments, we change the title to “Modulator effect of AT1 receptor knockdown in THP-1 macrophage proinflammatory activity”.
- “Caledon Laboratories, Canada” (lines 93-94) should be changed to “Caledon Laboratories, Chalk River, Canada”.
Answer: The text was changed
- “Na2HPO4” (line 96) should be changed to “Na<sub>2</sub>HPO<sub>4</sub>”.
Answer: The text was changed
- “KH2PO4” (line 96) should be changed to “KH<sub>2</sub>PO<sub>4</sub>”.
Answer: The text was changed
- “Sigma-Aldrich, EE. UU.” (lines 96-97) should be changed to “Sigma-Aldrich, Burlington, MA”.
Answer: The text was changed
- “25 cm2 flasks” (line 101) should be changed to “25 cm<sup>2</sup> flasks”.
Answer: The text was changed
- “5% CO2” (line 102) should be changed to “5% CO<sub>2</sub>”.
Answer: The text was changed
- “3 x 105 cells” (line 104) should be changed to “3 x 10<sup>5</sup> cells”.
Answer: The text was changed
9 . “2 x105 cells” (line 108) should be changed to “2 x10<sup>5</sup> cells”.
Answer: The text was changed
- “InvivoGen” (line 114) should be changed to “Invivogen”.
Answer: The word InvivoGen could not be change because is the name of the brand
- “2 x105 cells” (line 119) should be changed to “2 x10<sup>5</sup> cells”.
Answer: The text was changed
- In Table 1, “NF-KB” should be changed to “NF-κB”.
Answer: The text was changed
- “R&DSystems, Minneapolis, MN, USA” (lines 145-146) should be changed to “R&DSystems, Minneapolis, MN”.
Answer: The text was changed
- “Bio-Rad, Calif, USA” (line 148) should be changed to “Bio-Rad, Hercules, CA”.
Answer: The text was changed
- In Ref. 12, “Wang, C. H., Li, S. H., Weisel, R. D., Fedak, P. W., Dumont, A. S., Szmitko, P., ... & Verma, S.” should be changed to “Wang, C.H., Li, S.H., Weisel, R.D., Fedak, P.W., Dumont, A.S., Szmitko, P., Li, R.K., Mickle, D.A., Verma, S.”.
Answer: The text was changed
- In Ref. 13, “Recent Patents on Cardiovascular Drug Discovery (Discontinued), 2(1), 23-27” should be changed to “Recent Pat Cardiovasc Drug Discov. 2007;2(1):23-27.”.
Answer: The text was changed
- In Ref. 28, “Forrester, S. J., Booz, G. W., Sigmund, C. D., Coffman, T. M., Kawai, T., Rizzo, V., ... & Eguchi, S.” should be changed to “Forrester, S.J., Booz, G.W., Sigmund, C.D., Coffman, T.M., Kawai, T., Rizzo, V., Scalia, R., Eguchi, S.”.
Answer: The text was changed
- In Ref. 39, “Aki, K., Shimizu, A., Masuda, Y., Kuwahara, N., Arai, T., Ishikawa, A., ... & Fukuda, Y.” should be changed to “Aki, K., Shimizu, A., Masuda, Y., Kuwahara, N., Arai, T., Ishikawa, A., Fujita, E., Mii, A., Natori, Y., Fukunaga, Y., Fukuda, Y.”.
Answer: The text was changed
- In Ref. 40, “Journal of Diabetes Research, 2012” should be changed to “Exp. Diabetes Res. 2012;2012:618923.”.
Answer: The text was changed
- In Ref. 49, “Valsartan, independientemente del receptor AT1 o PPAR, suprime la activación de macrófagos inducida por LPS y mejora la resistencia a la insulina en adipocitos cocultivados.” should be changed to “Valsartan, independently of AT1 receptor or PPARγ, suppresses LPS-induced macrophage activation and improves insulin resistance in cocultured adipocytes.”.
Answer: The text was changed

Reviewer 2 Report
Comments and Suggestions for Authors
I got acquainted with the Acevedo-Villávicencio et al. manuscript submitted for review with interest. The work is carried out step by step, argumentatively, using the necessary experimental evidence of the effect of AT1R gene expression on the production of proinflammatory cytokines (TNF-α, IL-1β), as well as the relationship with blood pressure.
There are several suggestions for this work:
1. Eliminate spelling errors (for example, "al-ternative (alternative)", " renin angiotensin aldosterone (renin-angiotensin-aldosterone)").
2. A link to the source should be provided when specifying the GraphPad Prism 10 program used for calculations.
3. It seems important to discuss in the discussion what prospects for clinical use the data obtained by the authors may have.
4. It is recommended in the discussion to outline the prospects for further research on this topic with the potential practical use of the results obtained.
Author Response
Reviewer 2
- Eliminate spelling errors (for example, "al-ternative (alternative)", " renin angiotensin aldosterone (renin-angiotensin-aldosterone)").
Answer: the mistakes have been changed
- A link to the source should be provided when specifying the GraphPad Prism 10 program used for calculations.
Answer: The statistical analysis was performed using GraphPad Prism Version 7.0 (GraphPad Software, Inc., La Jolla, CA, USA)
- It seems important to discuss in the discussion what prospects for clinical use the data obtained by the authors may have.
Answer: The information was added in the discussion (line 386)
- It is recommended in the discussion to outline the prospects for further research on this topic with the potential practical use of the results obtained.
Answer: The information was added in the discussion (line 386)

Reviewer 3 Report
Comments and Suggestions for Authors
The manuscript Modulator effect of AT1 receptor knockdown in macrophage proinflammatory activity is very excellent work, well presented and well written. The work is of applied nature and results obtained are quite interesting. The authors have worked on the anti-inflammatory effects of AT1. The results obtained by the authors prove that AT1 receptor antagonist promotes the inhibition of the secretion of multiple proinflammatory cytokines in macrophages, as well as a decrease in the concentration of reactive oxygen species. The gene expression of IL-1β showed similar results in the group stimulated with Ang II for 1 h. The expression of this cytokine increased in the group stimulated with Ang II compared with the control group, these findings by the authors are quite interesting. Overall the quality of the manuscript is excellent and deserve for publication. I recommend this manuscript for publication after the incorporation of the following points
I suggest authors to rewrite the abstract to make it more clear and self-explanatory. While going through this interesting manuscript I find some old and ir-relevant references, through the literature i found some interesting and relevant work, i suggest authors to replace:
Replace reference number 1 with
Cao, J., Chen, C., Wang, Y., Chen, X., Chen, Z.,... Luo, X. (2016). Influence of autologous dendritic cells on cytokine‑induced killer cell proliferation, cell phenotype and antitumor activity in vitro. Oncol Lett, 12(3), 2033-2037. doi: 10.3892/ol.2016.4839
Replace reference 5 with
Yuqian Li, R. W. Q. G. (2023). The Roles and Targeting of Tumor-Associated Macrophages. FBL, 28(9), 207. doi: 10.31083/j.fbl2809207
Replace reference 24 with
Liang, W., Liu, H., Zeng, Z., Liang, Z., Xie, H., Li, W.,... Kang, L. (2023). KRT17 Promotes T-lymphocyte Infiltration Through the YTHDF2–CXCL10 Axis in Colorectal Cancer. Cancer Immunology Research, 11(7), 875-894. doi: 10.1158/2326-6066.CIR-22-0814
Replace reference 32 with
Mao, X., Chen, Y., Lu, X., Jin, S., Jiang, P., Deng, Z., Zhu, X., Cai, Q., Wu, C., & Kang, S. (2023). Tissue resident memory T cells are enriched and dysfunctional in effusion of patients with malignant tumor. Journal of Cancer, 14(7), 1223–1231. https://doi.org/10.7150/jca.83615
Replace reference 43 with
Wu, R., Xiong, J., Zhou, T., Zhang, Z., Huang, Z., Tian, S.,... Wang, Y. (2023). Quercetin/Anti-PD-1 Antibody Combination Therapy Regulates the Gut Microbiota, Impacts Macrophage Immunity and Reshapes the Hepatocellular Carcinoma Tumor Microenvironment. FBL, 28(12), 327. doi: 10.31083/j.fbl2812327
Incorporation of these interesting work can enhance the potential of the manuscript and eventually beneficial for the readers of the journal
Decision: Minor revision
Comments on the Quality of English Language
Minor editing of English language required
Author Response
Reviewer 3
I suggest authors to rewrite the abstract to make it more clear and self-explanatory. While going through this interesting manuscript I find some old and irrelevant references, through the literature i found some interesting and relevant work, i suggest authors to replace:
Answer: Thanks for the suggestions we changed the references
Replace reference number 1 with
Cao, J., Chen, C., Wang, Y., Chen, X., Chen, Z.,... Luo, X. (2016). Influence of autologous dendritic cells on cytokine‑induced killer cell proliferation, cell phenotype and antitumor activity in vitro. Oncol Lett, 12(3), 2033-2037. doi: 10.3892/ol.2016.4839
Replace reference 5 with
Yuqian Li, R. W. Q. G. (2023). The Roles and Targeting of Tumor-Associated Macrophages. FBL, 28(9), 207. doi: 10.31083/j.fbl2809207
Replace reference 24 with
Liang, W., Liu, H., Zeng, Z., Liang, Z., Xie, H., Li, W.,... Kang, L. (2023). KRT17 Promotes T-lymphocyte Infiltration Through the YTHDF2–CXCL10 Axis in Colorectal Cancer. Cancer Immunology Research, 11(7), 875-894. doi: 10.1158/2326-6066.CIR-22-0814
Replace reference 32 with
Mao, X., Chen, Y., Lu, X., Jin, S., Jiang, P., Deng, Z., Zhu, X., Cai, Q., Wu, C., & Kang, S. (2023). Tissue resident memory T cells are enriched and dysfunctional in effusion of patients with malignant tumor. Journal of Cancer, 14(7), 1223–1231. https://doi.org/10.7150/jca.83615
Replace reference 43 with
Wu, R., Xiong, J., Zhou, T., Zhang, Z., Huang, Z., Tian, S.,... Wang, Y. (2023). Quercetin/Anti-PD-1 Antibody Combination Therapy Regulates the Gut Microbiota, Impacts Macrophage Immunity and Reshapes the Hepatocellular Carcinoma Tumor Microenvironment. FBL, 28(12), 327. doi: 10.31083/j.fbl2812327
Incorporation of these interesting work can enhance the potential of the manuscript and eventually beneficial for the readers of the journal

Reviewer 4 Report
Comments and Suggestions for Authors
Dear Authors,
I have carefully reviewed your manuscript and would like to provide some feedback on the content:
1. There are two instances of "3.1" in the Results section. Please ensure consistency in numbering the subsections for clarity.
2. In paragraph 3.1, it is mentioned "As shown in Figure 2," which should be corrected to "As shown in Figure 1" to align with the correct figure reference.
3. It would be beneficial to provide a rationale for choosing the time points of 1 hour and 24 hours for the treatments in the study. Justification for these specific time intervals would enhance the understanding of the experimental design.
4. It appears that Figure 3 is missing from the manuscript. Please ensure that all figures are included and labeled correctly for a comprehensive presentation of the data.
5. There seems to be an error in the labeling of the groups in the figures. Please review and correct the group labels to accurately represent the experimental conditions.
6. It is recommended to provide a detailed explanation of the significance of the Δ symbol in the statistical analysis within the figures. Clarifying the meaning of this symbol will aid readers in interpreting the results accurately.
7. Consider expanding the analysis to include the detection of other inflammatory factors to provide a more comprehensive understanding of the impact of AT1 receptor inhibition on inflammation.
8. To enhance the credibility of the results, consider incorporating additional experiments such as Western blotting (WB) to validate the findings obtained through gene expression analysis.
Overall, the study shows promise, but addressing these points will strengthen the manuscript and improve its scientific rigor. Thank you for considering these suggestions for the enhancement of your research.
Best regards,
Comments on the Quality of English LanguageMinor editing of English language required according to my editor suggest.
Author Response
Reviewer 4
I have carefully reviewed your manuscript and would like to provide some feedback on the content:
Question
- There are two instances of "3.1" in the Results section. Please ensure consistency in numbering the subsections for clarity.
Answer: Thank you for your comments, your suggestions has been modified in the manuscript.
- In paragraph 3.1, it is mentioned "As shown in Figure 2," which should be corrected to "As shown in Figure 1" to align with the correct figure reference.
Answer: thank you for your comment, the text was change in the manuscript.
- It would be beneficial to provide a rationale for choosing the time points of 1 hour and 24 hours for the treatments in the study. Justification for these specific time intervals would enhance the understanding of the experimental design.
Answer: thank you for your comment. The experimental design at 1 hour of stimulation with Ang II was developed to evaluate the immediate responses in the activation of macrophages, while the evaluation at 24 hours was carried out to analyse the modulation in the release of proinflammatory cytokines. the pathogenesis of cardiovascular diseases (hypertensive cardiac remodeling), the Inflammation plays an important role in [3] because the Angiotensin (Ang) II, mediates early infiltration of pro-inflammatory cells (monocytes/macrophages) in this affection.
McMaster WG, Kirabo A, Madhur MS, Harrison DG. Inflammation, immunity, and hypertensive end-organ damage. Circ Res. 2015;116:1022–33.
Zhu YC, Zhu YZ, Lu N, Wang MJ, Wang YX, Yao T. Role of angiotensin AT1 and AT2 receptors in cardiac hypertrophy and cardiac remodelling. Clin Exp Pharmacol Physiol. 2003;30:911–8.
O’Rourke SA, Dunne A, Monaghan MG. The role of macrophages in the infarcted myocardium: orchestrators of ECM remodeling. Front Cardiovasc Med. 2019;6:101.
Jia L, Li Y, Xiao C, Du J. Angiotensin II induces inflammation leading to cardiac remodeling. Front Biosci (Landmark Ed) 2012;17:221–31.
- It appears that Figure 3 is missing from the manuscript. Please ensure that all figures are included and labeled correctly for a comprehensive presentation of the data.
Answer: thank you for your comment, the figure has been added in the manuscript.
- There seems to be an error in the labeling of the groups in the figures. Please review and correct the group labels to accurately represent the experimental conditions.
Answer: Thank you for your comments: the changes were done in the figures.
- It is recommended to provide a detailed explanation of the significance of the Δ symbol in the statistical analysis within the figures. Clarifying the meaning of this symbol will aid readers in interpreting the results accurately.
Answer: Thank you for your comments: the changes were done in the manuscript.
- Consider expanding the analysis to include the detection of other inflammatory factors to provide a more comprehensive understanding of the impact of AT1 receptor inhibition on inflammation.
Answer: It's an interesting question, we appreciate your comment, our future perspective for future studies is to analyze the signaling cascade of the AT2 receptor, since it could be playing a relevant role in inflammation, because AT1 and AT2 self-regulate.
- To enhance the credibility of the results, consider incorporating additional experiments such as Western blotting (WB) to validate the findings obtained through gene expression analysis.
Answer: thank you for your comments. We made the decision to measure protein release of macrophages using the ELISA method. We will consider the measurement of intracellular proteins measure out in future research, using the Western blot technique.

Round 2
Reviewer 1 Report
Comments and Suggestions for Authors
Most of points were suitably revised in biology-2987814-v2. A few points to be re-considered/changed (pointed out in <Comments to Authors>) for the benefit of readers. These are given below.
<Point>
1. In Figure 3A & its legend, the authors described, “A) IL-1β gene expression in macrophages stimulated with Ang II (1µM) for 1 h” in legend (lines 280-281). However, the Figure 3A showed “Fold change in mRNA expression TNF-α/GAPDH” (1 h).
2. In Figure 3B & its legend, the authors described, “B) IL-1β gene expression in macrophages stimulated with Ang II (1µM) for 24 h” (lines 281-282). However, the Figure 3B showed “Fold change in mRNA expression TNF-α/GAPDH” (24 h).
3. In Figure 3C & its legend, the authors described, “C) TNF-α gene expression in macrophages stimulated with Ang II (1µM) for 1 h” (line 282). However, the Figure 3C showed “Fold change in mRNA expression IL-1β/GAPDH” (1 h).
4. In Figure 3D & its legend, the authors described, “D) TNF-α gene expression in macrophages stimulated with Ang II (1µM) for 24 h” (lines 282-283). However, the Figure 3D showed “Fold change in mRNA expression IL-1β/GAPDH” (24 h).
5. In Figure 3E & its legend, “E) IL-10 gene expression in macrophages stimulated with Ang II (1 µM) for 1 h” (lines 283-284). However, there is no Figure 3E.
6. In Figure 3F & its legend, “F) IL-10 gene expression in macrophages stimulated with Ang II (1µM) for 24 h.” (lines 284-285). However, there is no Figure 3F.
7. “HyClone, USA” (line 111) should be changed to “HyClone, Logan, UT”.
8. “The absorbance value (OD 450 nm) was measure” (line 179) should be changed to “The absorbance value (OD 450 nm) was measured”.
9. In Ref. 5, “FBL, 28(9), 207” should be changed to “Front. Biosci. (Landmark Ed) 2023, 28, 207”.
10. In Ref. 13, ““Macrophage polarization and function with emphasis on the evolving roles of coordinated regulation of cellular signaling pathways,” Cell. Signal., vol. 26, no. 2, pp. 192–197, 2014” should be changed to “Macrophage polarization and function with emphasis on the evolving roles of coordinated regulation of cellular signaling pathways. Cell Signal. 2014, 26, 192-197.”.
11. In Ref. 14, ““Role of renin-angiotensin system in activation of macrophages by modified lipoproteins,” Am. J. Physiol. - Hear. Circ. Physiol., vol. 305, no. 9, pp. 1309–1320, 2013” should be changed to “Role of renin-angiotensin system in activation of macrophages by modified lipoproteins. Am. J. Physiol. Heart Circ. Physiol. 2013, 305, H1309-H320.”.
12. In Ref. 16, ““Role of angiotensin II type 1 receptor in angiotensin II-induced cytokine production in macrophages,” J. Interf. Cytokine Res., vol. 31, no. 4, pp. 351–361, 2011” should be changed to “Role of angiotensin II type 1 receptor in angiotensin II-induced cytokine production in macrophages. J. Interferon Cytokine Res. 2011, 31, 351-361.”.
13. In Ref. 23, ““Precise and efficient siRNA design: A key point in competent gene silencing,” Cancer Gene Ther., vol. 23, no. 4, pp. 73–82, 2016” should be changed to “Precise and efficient siRNA design: a key point in competent gene silencing. Cancer Gene Ther. 2016, 23, 73-82.”.
14. In Ref. 31, “Macrophages and NF-κB in cancer. NF-kB in Health and Disease, 171-184” should be changed to “Macrophages and NF-κB in cancer. Curr. Top. Microbiol. Immunol. 2011, 349, 171-184.”.
15. In Ref. 33, “La regulación del factor de transcripción NF-κB. Un mediador molecular en el proceso inflamatorio. Revista de investigación clínica, 56(1), 83-92” should be changed to “La regulación del factor de transcripción NF-κB. Un mediador molecular en el proceso inflamatorio [Regulation of NF-κB transcription factor. A molecular mediator in inflammatory process]. Rev. Invest. Clin. 2004, 56, 83-92. Spanish”.
16. In Ref. 36, “"Proinflammatory actions of angiotensins." Current opinion in nephrology and hypertension 10.3 (2001): 321-329” should be changed to “Proinflammatory actions of angiotensins. Curr. Opin. Nephrol. Hypertens. 2001, 10, 321-329.”.
17. In Ref. 43, “FBL” should be changed to “Front. Biosci. (Landmark Ed).”.
Author Response
Reviewer 1
- In Figure 3A & its legend, the authors described, “A) IL-1β gene expression in macrophages stimulated with Ang II (1µM) for 1 h” in legend (lines 280-281). However, the Figure 3A showed “Fold change in mRNA expression TNF-α/GAPDH” (1 h).
Answer: thank you for your comment, the text has been modified in legend of the figure 3 A).
- In Figure 3B & its legend, the authors described, “B) IL-1β gene expression in macrophages stimulated with Ang II (1µM) for 24 h” (lines 281-282). However, the Figure 3B showed “Fold change in mRNA expression TNF-α/GAPDH” (24 h).
Answer: thank you for your observation, the text has been modified in legend of the figure 3 B).
- In Figure 3C & its legend, the authors described, “C) TNF-α gene expression in macrophages stimulated with Ang II (1µM) for 1 h” (line 282). However, the Figure 3C showed “Fold change in mRNA expression IL-1β/GAPDH” (1 h).
Answer: the text has been modified in legend of the figure 3 C).
- In Figure 3D & its legend, the authors described, “D) TNF-α gene expression in macrophages stimulated with Ang II (1µM) for 24 h” (lines 282-283). However, the Figure 3D showed “Fold change in mRNA expression IL-1β/GAPDH” (24 h).
Answer: the text has been modified in legend of the figure 3 D).
- In Figure 3E & its legend,“E) IL-10 gene expression in macrophages stimulated with Ang II (1 µM) for 1 h” (lines 283-284). However, there is no Figure 3E.
Answer: thank you for your comment, the figure has been added in the manuscript.
- In Figure 3F & its legend,“F) IL-10 gene expression in macrophages stimulated with Ang II (1µM) for 24 h.” (lines 284-285). However, there is no Figure 3F.
Answer: the figure has been added in the manuscript.
- “HyClone, USA” (line 111) should be changed to “HyClone, Logan, UT”.
Answer: the text has been changed
- “The absorbance value (OD 450 nm) was measure” (line 179) should be changed to “The absorbance value (OD 450 nm) was measured”.
Answer: the text has been changed
- In Ref. 5, “FBL, 28(9), 207” should be changed to “Front. Biosci. (Landmark Ed) 2023, 28, 207”.
Answer: the text has been changed
- In Ref. 13, ““Macrophage polarization and function with emphasis on the evolving roles of coordinated regulation of cellular signaling pathways,” Cell. Signal., vol. 26, no. 2, pp. 192–197, 2014” should be changed to “Macrophage polarization and function with emphasis on the evolving roles of coordinated regulation of cellular signaling pathways. Cell Signal. 2014, 26, 192-197.”.
Answer: the text has been changed
- In Ref. 14, ““Role of renin-angiotensin system in activation of macrophages by modified lipoproteins,” Am. J. Physiol. - Hear. Circ. Physiol., vol. 305, no. 9, pp. 1309–1320, 2013” should be changed to “Role of renin-angiotensin system in activation of macrophages by modified lipoproteins. Am. J. Physiol. Heart Circ. Physiol. 2013, 305, H1309-H320.”.
Answer: the text has been changed
- In Ref. 16, ““Role of angiotensin II type 1 receptor in angiotensin II-induced cytokine production in macrophages,” J. Interf. Cytokine Res., vol. 31, no. 4, pp. 351–361, 2011” should be changed to “Role of angiotensin II type 1 receptor in angiotensin II-induced cytokine production in macrophages. J. Interferon Cytokine Res. 2011, 31, 351-361.”.
Answer: the text has been changed
- In Ref. 23, ““Precise and efficient siRNA design: A key point in competent gene silencing,” Cancer Gene Ther., vol. 23, no. 4, pp. 73–82, 2016” should be changed to “Precise and efficient siRNA design: a key point in competent gene silencing. Cancer Gene Ther. 2016, 23, 73-82.”.
Answer: the text has been changed
- In Ref. 31, “Macrophages and NF-κB in cancer. NF-kB in Health and Disease, 171-184” should be changed to “Macrophages and NF-κB in cancer. Curr. Top. Microbiol. Immunol. 2011, 349, 171-184.”.
Answer: the text has been changed
- In Ref. 33, “La regulación del factor de transcripción NF-κB. Un mediador molecular en el proceso inflamatorio. Revista de investigación clínica, 56(1), 83-92” should be changed to “La regulación del factor de transcripción NF-κB. Un mediador molecular en el proceso inflamatorio [Regulation of NF-κB transcription factor. A molecular mediator in inflammatory process]. Rev. Invest. Clin. 2004, 56, 83-92. Spanish”.
Answer: the text has been changed
- In Ref. 36, “"Proinflammatory actions of angiotensins." Current opinion in nephrology and hypertension 10.3 (2001): 321-329” should be changed to “Proinflammatory actions of angiotensins. Curr. Opin. Nephrol. Hypertens. 2001, 10, 321-329.”.
Answer: the text has been changed
- In Ref. 43, “FBL” should be changed to “Front. Biosci. (Landmark Ed).”.
Answer: the text has been changed

Reviewer 4 Report
Comments and Suggestions for Authors
Dear Authors,
I have carefully reviewed the revised version of your manuscript titled "Effect of AT1 Receptor Gene Silencing on Cytokine Modulation in THP-1 Macrophages" and while I appreciate the efforts made in addressing the previous concerns, there are still some outstanding issues that need to be resolved:
1. Figure 3 is still missing from the manuscript. Including this figure is essential for a complete understanding of the results presented in the study.
2. In Figure 4, the labeling of the subfigures should be corrected to indicate as A and B for clarity and consistency in the presentation of the data.
I kindly request that you address these remaining issues in the manuscript before finalizing it for publication. Once these adjustments are made, the manuscript will be closer to meeting the publication standards.
Thank you for your attention to these matters, and I look forward to seeing the revised manuscript with the necessary corrections.
Best regards,
Author Response
Reviewer 2
- Figure 3 is still missing from the manuscript. Including this figure is essential for a complete understanding of the results presented in the study.
Answer: thank you for your comment, the figure has been added in the manuscript (figure 3 E).
- In Figure 4, the labeling of the subfigures should be corrected to indicate as A and B for clarity and consistency in the presentation of the data.
Answer: thank you for your comment, the figure 4 has been modified.

Round 3
Reviewer 4 Report
Comments and Suggestions for Authors
Dear Authors,
I have completed the second round of review for your manuscript titled "Modulatory Effect of AT1 Receptor Inhibition on Proinflammatory Activity of THP-1 Macrophages". I am pleased to inform you that all the issues I previously raised have been successfully addressed in the revised version of the manuscript. I appreciate your attention to detail and the thorough revisions made in response to the feedback provided.
The clarity of the objectives, the detailed methodology descriptions, and the organization of the results have been significantly improved in the revised manuscript. The changes made have enhanced the overall quality of the study and have effectively addressed the concerns raised during the first round of review.
Thank you for your hard work and dedication to improving the manuscript. I look forward to seeing your research published.
Best regards,